# Advance Care Planning (ACP) vs. Advance Serious Illness Preparations and Planning (ASIPP)

**DOI:** 10.3390/healthcare8030218

**Published:** 2020-07-18

**Authors:** Daren K. Heyland

**Affiliations:** 1Department of Critical Care Medicine, Queen’s University, Kingston, ON K7L 2V7, Canada; dkh2@queensu.ca; Tel.: +1-(403)-915-5573; 2Clinical Evaluation Research Unit, Kingston Health Science Centre, Kingston, ON K7L 2V7, Canada

**Keywords:** end of life, serious illness, advance care planning, communication and decision-making, critical care

## Abstract

COVID-19 has highlighted the reality of an impending serious illness for many, particularly for older persons. Those faced with severe COVID-19 infection or other serious illness will be faced with decisions regarding admission to intensive care and use of mechanical ventilation. Past research has documented substantial medical errors regarding the use or non-use of life-sustaining treatments in older persons. While some experts advocate that advance care planning may be a solution to the problem, I argue that the prevailing understanding and current practice of advance care planning perpetuates the problem and results in patients not receiving optimal patient-centered care. Much of the problem centers on the framing of advance care planning around end of life care, the lack of use of decision support tools, and inadequate language that does not support shared decision-making. I posit that a new approach and new terminology is needed. Advance Serious Illness Preparations and Planning (ASIPP) consists of discrete steps using evidence-based tools to prepare people for future clinical decision-making in the context of shared decision-making and informed consent. Existing tools to support this approach have been developed and validated. Further dissemination of these tools is warranted.

## 1. Introduction

With COVID-19 having spread around the world at such an alarming rate, the impact this serious illness might have on people and their loved ones caused much anxiety. To reduce this stress, some people were calling for end of life or Advance Care Plans (ACP) [1,2]. Traditionally, these ACPs have focused on end-of-life or terminal care plans. Unfortunately, those plans are not necessarily suitable for the situation we are currently faced with as a result of COVID-19; planning for death under conditions of certainty (like when you have end-stage cancer) is not the same as planning for serious illnesses with uncertain outcomes (like COVID-19 pneumonia). Let me explain.

I remember talking to an older lady in her home about her advance medical care plans. Immediately, when we started the conversation about her wishes, she emphatically stated, “I would never want to be on machines” as she shook her head with a firm look of determination on her face. I was sitting in her living room in a nicely appointed, well-kept house. She was a spry, fit older person. It seemed odd to me that she should categorically refuse such potentially life-sustaining treatments. I asked, “So, what if, when you were sick there was a probability of recovery in which we could get you better and back to your baseline function, would it be worth it to you go (sic) on machines temporarily?” The previous look of determination vanished and gave way to a wide-eyed, almost panicked look as she said, “I don’t know—what do you think?” I went on to explain the nature of serious illness, that we do not always know if patients will recover or die, and if they recover, what kind of shape the person will be in. She later confessed that she thought we were talking about her end of life, that when she was dying she did not want to have her dying experience prolonged with machines.

This real-life anecdote highlights the complexity and challenge of communication and decision-making in the context of serious illness. Many people and health care professionals associate advance care planning (ACP) with planning for future end of life or terminal care [3,4,5,6]. This movement stems from society’s understandable fear regarding how technology unleashed can keep people alive in health state’s worse than death. Consequently, in many states and provinces, people are encouraged to fill out advance directives or living wills that pre-specify the medical treatments they want or will not want when they are dying. ACP, advance directives (AD), living wills, etc., have been embraced by palliative care clinicians, dying with dignity societies, and hospice palliative care organizations. In Canada, the advance care planning organization is situated in the Canadian Hospice and Palliative Care Association, for example. Whilst all of this effort is seemingly good, is it plausible that it could be causing harm or at least confusion within the walls of the medical–industrial complex and even with society at large?

## 2. The Problems with Making Treatment Decisions in Advance

Most of the care plans or instruction directives are framed around conditions of certainty, “If I am dying, I don’t want this or I do want this....” However, most of the clinical scenarios where we need to make preference-sensitive decisions regarding the use of life-sustaining treatments are situated early in the clinical course where the outcomes are uncertain. I remember a patient who was intubated with respiratory failure secondary to heart failure and transferred up to my ICU. When I met with the family, they produced a living will that said he did not want any “heroics”, and the family was upset that his wishes were not honored. I set aside the written document to explore what they thought he meant when he wrote the document. Once again, they confirmed that he was trying to say that when he was dying or in some persistent terrible health state, such as coma, that he would not want life-sustaining machines. I asked, “When he made this living will, do you think he was thinking of a situation like this, where with just a few hours of positive pressure ventilation and diuresis, we have a good chance of getting him off the machines quickly and fully recovered? They unanimously agreed he was likely not thinking of his current clinical context when he filled out his living will. As a critical care doctor, I would argue in the context of life and death decision-making, we cannot have uncertainty, misunderstandings, or ambiguity about the meaning of people’s wishes. So, what then is the value of these kinds of ACP, AD, and/or living wills in the context of serious illness? They may be helpful in planning your death (under conditions of certainty), but they are not helpful in the context of serious illness where the outcome is uncertain when decisions have to be made.

Furthermore, does it seem sensible that a lay person sitting at their kitchen table or even with a lawyer in their office should make medical treatment decisions when they are ill-informed about medical treatments and more likely driven by emotion and fear with no one to coach or help them deal effectively with their future realities? Gianluca Montanari Vergallo recently argued eloquently why advance directives or living wills should be considered “invalid” (except in specific, limited circumstances, which I will explain later) [7]. For a summary of his reasoning, see Box 1.
Box 1Reasons Advance Directives or Living Wills Are Not Useful.Advance decisions have no situational validity
○Wishes expressed in an AD may not be reflective of the situation encountered in the future disease. For example, Patient in respiratory failure secondary to heart failure with a living will rejecting the use of artificial life supports.Person cannot be fully informed about treatment options, risks, and benefits as required by health care law.
○With the passage of time, new developments may have occurred both to the person and to the medical treatment optionsLanguage too vague and non-specific to be clinically useful
○What do the following terms mean: “Hopeless,” “artificial” vs “natural, etc.?Advance decisions are unreliable
○There is no certainty that persons who made decisions in advance would confirm the same decision when facing the actual disease.Incapable patients cannot revoke their decisions (whereas capable patients have that right)

I think this “lack of validity and reliability” of said “death plans” explain findings from recent published studies of ACP. In a multicenter audit of “Physician Order for the use of Life-Sustaining Treatments (POLST)” documents published in JAMA [8], Lee and colleagues analyzed the health care trajectory of patients living with chronic illness who had a POLST and were hospitalized in the last 6 months of life. They documented the impact of POLST on end of life care that up to 38% of patients with POLST receive “discordant” care. These were mostly patients that had signed up for comfort measures only or some other limitation on the use of life supports and yet were admitted to ICU and received life-sustaining treatments. While acknowledging the complexity of clinical decision-making, the authors suggest that discordant care may be unwanted and that we are doing harm (not to mention wasting resources) by providing intensive care services to such patients who did not want them in the first place. I posit that an alternate explanation may be that POLST/Advance Directives/Living wills, or in other words, specifying medical treatment decisions in advance is a flawed paradigm that uses flawed tools. Critical care physicians do not have crystal balls, and at the point they are making decisions about ICU admission and intubation, they do not know the patient is dying, and hence, the patient’s death plans are not followed.

A qualitative study from Australia offers further validation of this concern [9]. These investigators performed a qualitative study of doctor’s attitudes and beliefs about advance care planning. While the authors of the JAMA paper on POLST want to encourage doctors to pay more attention to the previously documented wishes of patients, many doctors looking after these critically ill patients are suspicious about the validity of the prior-expressed wishes (see Box 2 for illustrative quotes from this study) [9].
Box 2Doctor’s Concerns About the Validity of Advance Directives.“I guess with any documentation, you’re presuming it was done under the right circumstances, without coercion and all of that … We can never verify that unless we were part of the process”.  (Physician).“It’s also useful to say, “I don’t want to be intubated”, but it’s kind of a harder one for some patients, who haven’t had much experience with hospitals, who haven’t had medical training, to kind of understand that. So, I think, it’s probably, more for a patient-centered approach, “I want a quality of life” or “I don’t want to be in a nursing home”, because they can definitely understand that … If they happen to be a nurse or a doctor, then that’s fine. If they haven’t had much medical experience or medical training, you’d want to know, well, what is it about intubation you don’t like? Or, what have you heard? Their understanding of intubation might be very different to my understanding of intubation”. (Infectious disease and general medicine consultant).“The advance care plan that was made 5 years ago could potentially be outdated, especially with the potentially new diagnoses of metastatic pancreatic cancer and end-stage kidney disease, that could very well change the patient’s perspective about what he wants to do about limitations of treatment”. (Nephrology consultant).

The key point is that planning for death under conditions of certainty (like when you have end-stage cancer) is not the same as planning for your serious illness (like COVID-19 pneumonia) where it is uncertain how you will turn out. This is further illustrated by a recent audit of patients with advanced cancer who also had multiple comorbidities, like congestive heart failure, chronic pulmonary disease, dementia, etc. [10]. Compared to patients with no or few chronic conditions, patients with multiple chronic conditions experience increased intensity of care at the end of life. In particular, cancer patients with heart failure had highest odds of hospitalization (odds ratio (OR) 1.67, 95% confidence interval (CI) 1.46–1.91), ICU admission (OR 2.06, 95% CI 1.76–2.41), or in-hospital death (OR 1.62, 95% CI 1.43–1.84) compared to patients with cancer and other comorbidities. Why might this be? When a patient with cancer and heart failure presents to the emergency room short of breath, even with an advance directive indicating comfort measures, treating physicians do not know if this is truly the end of life. There still is a possibility that this patient with an incurable cancer but a curable acute condition recovers from this episode of illness, so medical treatments are escalated, and care is intensified.

Another reason that current approaches to ACP may not be trustworthy relates to the way in which values are elicited and translated into medical orders for the use of life-sustaining treatments. We have previously shown eliciting values in an open-ended, unconstrained manner, like what often happens in the real world, whereby the patient does not explicitly see the conflict between competing values may not be helpful in determining the best plan of care for seriously ill patients [11]. Lay people’s expressed values often conflict with each other and bear little relationship to their preferences for medical care [11]. When a person states, “My mom is a ‘fighter’!”, they imply that she should be given every chance at curative treatment. What is not made transparent is the consequence of this value-driven choice that survival comes at a cost of impaired quality of life. Furthermore, this research conducted by our group found a considerable disconnect between older patients’ stated values and their preferences related to the use of life-sustaining treatments, which makes the interpretation of their values problematic [11]. Moving forward, eliciting values from lay people needs to use constraining values elicitation tools, something that is NOT part of the current ACP approach.

In addition, physicians tend to treat these patients as “informed consumers” asking them “what they want”, and our research documents a profound lack of knowledge and understanding regarding CPR resuscitation, a key medical decision for hospitalized patients [12]. These findings challenge the foundation of advance care planning, which is that patients are informed consumers and can articulate authentic values and valid preferences that guide clinicians to writing medical orders. Moreover, the translation from POLST to medical orders for the use or non-use of life-sustaining treatments is irreproducible and error prone [13,14].

## 3. A New Approach: Advance Serious Illness Preparations and Planning (ASIPP)

As a strong advocate for ACP, I have come to the conclusion that we need to move away from the current approaches and terminology. ACP, as currently defined, is too broad and imprecise. International consensus definitions were recently adjusted and meant to encompass all “future” medical decision-making, which bundles together end of life decisions (under conditions of certainty) with serious illness (under conditions of uncertainty) [15]. ACP is widely misunderstood or not understood by lay people (even though they are engaging in ACP activities such as naming a substitute decision-maker) [6,16]. Nor is it really embraced and “trusted” by health care professionals. So, if the term ACP is not helpful to the key stakeholders that are meant to be using it, may be its time for a change? 

But to what? How can we still help people have some say or some control in the medical care they receive, particularly when they are so sick, they cannot participate in clinical decision-making. What can we do to help people prepare for future periods of incapacitation when their “voice” cannot be expressed? If we move away from “planning for the end of life” and begin to help people prepare to confront future periods of incapacitation, we begin to see the way forward. From the clinical point of view, we as doctors are obligated to make decisions according to health care laws that say we must fully explain the treatment options, and the risks, benefits, and possible outcomes of those options to our patients in the moment when treatment decisions need to be made. In those moments, we need patients to be able to express their authentic values and informed treatment preferences so, when combined with illumination from the physician regarding the clinical disease, the prognosis, and possible treatment options, the patient and the physician can collaborate in what we refer to as a “shared decision-making” (see Figure 1) approach to make the decision that is best for the patient. 

For shared decision-making to work then, patients need to be able to express their authentic values and informed treatment preferences. Our prior research has shown that lay people struggle to do this and that their lack of “being prepared” for this kind of shared decision-making is a major barrier for physicians to engage patients in open communication and optional decision-making [11,12,17]. Thus, our efforts should be focused on preparing people for future conversations with physicians so when they are seriously ill, they can effectively participate as partners and make good decisions in the moment. Unfortunately, when most people are seriously ill and this kind of life or death decision-making is required, people are incapacitated or incapable of speaking for themselves [18]. So, an important part of preparing for future periods of incapacitation is the naming and qualifying of an individual to make treatment decisions for you. The “naming” of a substitute decision maker (the person to make decisions for you when are your rendered uncapable) is part of the current advance care planning/instructional directive movement. But I argue not enough attention is paid to capacitating the substitute decision-maker to accurately speak on behalf of the patient, so they can voice the authentic values and preferences of the person. There needs to be standardized tools and language strategies promoted by clinicians and lawyers that effectively accomplish this goal.

So, now we come to the bottom line: Efforts need to focus on preparing people and their surrogates for future periods of incapacitation due to illness or injury, so they (or their surrogates) can be decisionally ready to engage with the doctors in shared decision-making about the medical treatments that are “right” for that patient given the specific clinical problem. It has been more than a decade since leaders in the field proposed this concept of preparing patients for future decision-making, but it was still ensconced in the context of end of life medical care [19]. Over the past decade, have we made progress in “preparing patients for end of life decision-making?” Recent studies suggest there are still considerable problems with the current approach [6,7,8,9,10,11,12,13,14,16]. Health care expenditures in the last years of life are increasing at a phenomenal rate suggesting the current approach to “end of life” planning is not working [20]. Would reframing this around “preparing and planning for serious illness” be helpful? Would there be greater engagement and clarity of role and process by both public and professionals by the reframing around serious illness versus just trying to improve the clarity and process of optimal preparations and planning under the “old” ACP banner? I would like to propose we rally around the concept of “Advance Serious Illness Preparation and Planning (ASIPP).” Ideally, before a crisis, ASIPP calls for asking patients their values in a way that highlights the trade-off with competing values. Questions like, “Are you the kind of person that wants medical treatments to focus on prolonging your life or enhancing the quality of your remaining days?” and “Are you the kind of person who prefers a natural death or are you willing to accept the use of machines, such as breathing machines, to prolong your life, for as long as possible?” The answers to these questions allow doctors to link stated values to medical treatments that could be proposed to treat serious illness in a reliable and transparent way, thus reducing medical errors (see Figure 2) [21]. As part of serious ill preparations, I suggest more systematic use of decision aids, such as the Plan Well Guide, that are useful in informing patients about the risks, benefits, and possible outcomes of these different treatment options highlighted in the Figure 2. Once informed, asking patients, “Are you willing to put up with the risks and possible outcomes of critical care treatments?” will help doctors then propose the “right” treatment plan for the seriously ill patient using the language of shared decision-making. In a recent randomized trial, this approach to serious illness planning was shown to improve decisional quality, reduce physician time and resulted in both physician and patient ratings of satisfaction and endorsement [21].

The one exception to the above focus on “preparing”, which I allude to above when citing Vergallo’s writings, is when a patient living with an advanced chronic disease or life-limiting illness, wants to make a specific treatment plan for future medical care with their doctor. The difference here to prior comments about instructional advance directives is that these directives are specific to the present disease, are based on the patients lived experience/knowledge of the disease, are made in conjunction with the treating clinical team, and are documented in the “forms” used by the local health care system (see Figure 3). I have provided some clinical examples of what I mean in the Box 3. These are examples of “informed” treatment decisions made in advance to forgo life-sustaining treatments and preserve dignity at the end of life made by patients experienced with the disease and in conjunction with relevant doctors. This kind of “advance directive” seems legitimate.
Box 3Clinical Examples of Where Advance Directives May be Legitimate.A patient with severe chronic obstructive lung disease who has had frequent exacerbations and even past intubations and trips to the ICU who, in discussions with their treating physician, makes a future decision to forgo further attempts at invasive mechanical ventilation. A patient with advanced cancer who has undergone several rounds of chemotherapy and requests to stop all future curative treatments, not to have CPR, and requests transfer to hospice.A patient with early dementia, while still capable, documents their wishes to not receive any life-prolonging treatments including artificial nutrition at later stages of their disease.

## 4. Conclusions

In summary, the problem is that much of the current ACP/AD movement has people (not patients) doing end of life planning/medical decision-making under conditions of certainty, devoid of context, and without support of clinical input. I propose that the way forward is to better prepare people for future serious illness decision-making, so they (or their surrogates) can better articulate their authentic values and informed treatment preferences in the context of “in the moment” shared medical decision-making with doctors AND to help seriously ill patients with capacity to better plan their future medical care with their treating clinicians on documents recognized by the health care system (ASIPP) (See Figure 3). Decision aids that help patients understand the context of serious illness and making decisions under conditions of uncertainty; help patients clarify their authentic values using constraining values clarification tools; and provide information in the form of a decision aid regarding the differences between ICU medicine, medical care, and comfort care are needed to support this new approach. Early versions exist and have been demonstrated to be acceptable and effective (www.planwellguide.com and www.myicuguide.ca) but are not widely disseminated [21,22]. It’s time for a change! People need to do their ASIPP ASAP!

## Figures and Tables

**Figure 1 healthcare-08-00218-f001:**
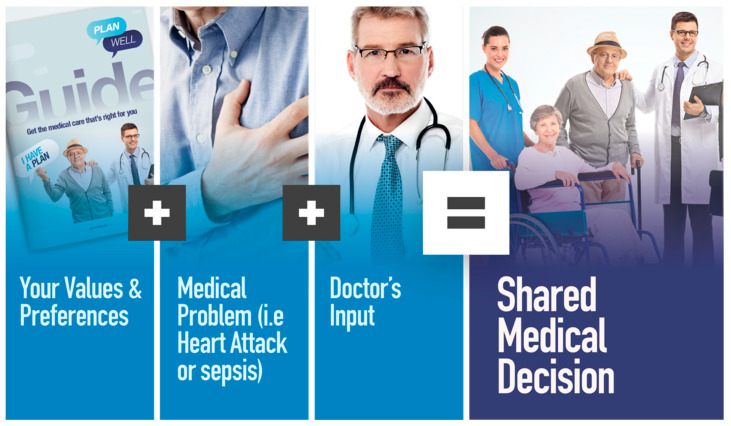
Shared decision-making. Caption: Shared decision-making is a combination of patient values and preferences and the doctor’s input in the context of the medical problem (a stock image used with permission from shutter stock).

**Figure 2 healthcare-08-00218-f002:**
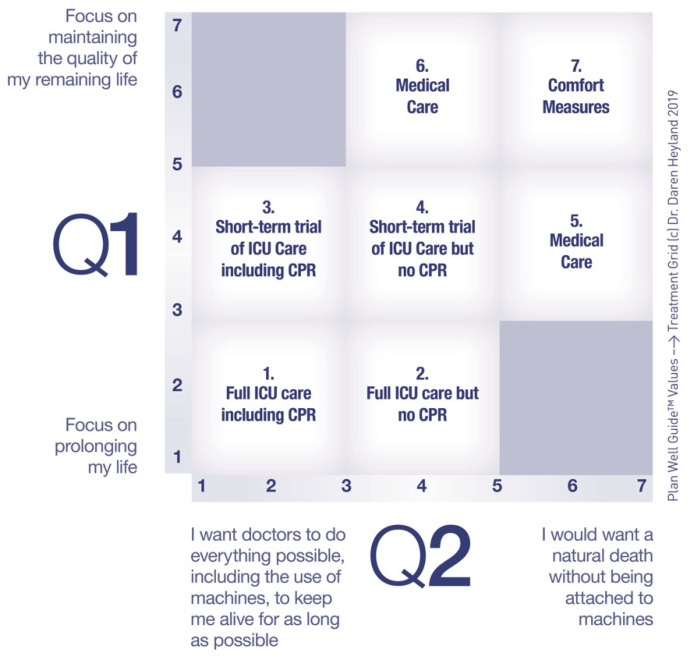
Values–Preferences Grid: By lining up the answers to the constraining values clarification scales on the X and Y axis, the intersection of the responses to the values questions leads to a cell that might be preferred by the patient. In this way, clinical decision-making is more reliable and transparent.

**Figure 3 healthcare-08-00218-f003:**
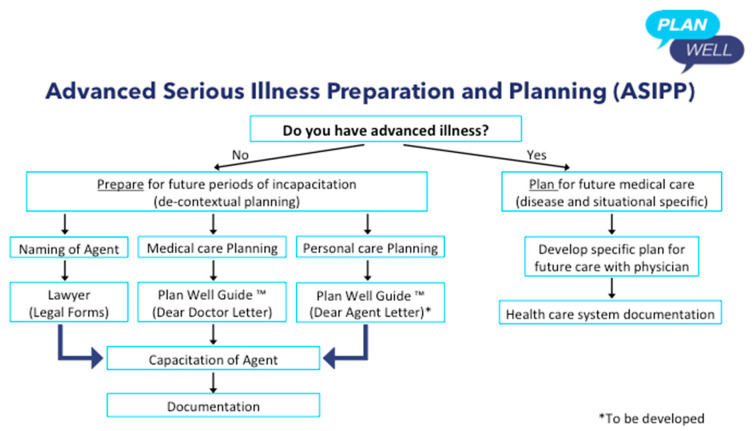
Advance Serious Illness Preparation and Planning. Caption: Flow diagram for ASIPP. Preparing consists of thinking ahead and planning ahead for future illness (Decontextualized planning) while “Planning” consists of preparing medical treatment orders for future care around a specific, known illness and illness trajectory using medical order forms specific to the institution or health care system.

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
