# Peer review of "Advance Care Planning (ACP) vs. Advance Serious Illness Preparations and Planning (ASIPP)"

_healthcare, 2020, doi:10.3390/healthcare8030218_

Round 1
Reviewer 1 Report
On a number of grounds, the author expresses scepticism about the value, and even the validity, of advanced care planning. His particular concern is that communication between patient and doctor is rarely good enough to explore or even to set out clearly what the areas of uncertainty are. The result, he argues, is that patients wrongly assume that any advance decisions they make relate only to a time when it is certain that death is an imminent outcome.
I have sympathy with the author's concerns, but I do not think that in this piece he sets them out clearly enough or justifies them convincingly. The language is informal and often vague and the argument is not coherent.
At times it appears to be an ethical argument, but it is not clear what fundamental principles he is appealing to. He refers to 'a right to self-determination' but it is not clear what the basis of that right is. In the context of medical decisions a right to self-determination is quite restricted; a patient can refuse an intervention she doesn't want, but ethically (and, in most jurisdictions, legally) she can't insist on one she does want unless it is justifiable in some other way; usually by the benefit it will offer vs. its harm.
The author also refers to a 'waste of resources' but doesn't define what he means. Is a resource wasted because it is spent on something a patient doesn't want, even if it will help her, or because it is spent on something she does want but will in fact harm her ?
The author's argument is strongest, it seems to me, when he appeals to the practical difficulty of exploring what may or may not happen in the future as a matter of probability. Here he is on solid ground. To be rationally arrived at, a categorical decision must be made by a person with capacity who has all the relevant facts at her disposal. Those relevant facts cannot usually be known with certainty before the event occurs that requires the decision. The only rational decisions that can therefore be made ahead of time are hypothetical ones "If this were to happen, then this is what I would want, whereas if that were to happen, that is what I would want... ' The author correctly points out that such conversations demand sensitive and sophisticated communication skills and need to take place early enough in the course of a patient's illness that she can explore all those possibilities and new ones as and when circumstances change. I am not familiar with the ASIPP tool the author refers to but on the basis of what this article says about it would agree that it supports such careful communication. In passing, this approach is the usual one in palliative care in children, where incurable conditions are often diagnosed many years or decades before death.
I would suggest the author re-write the piece, making it much more concise and clearer. That would be helped by focusing on the issue of communication rather than invoking ethics and citing more of the relevant empirical evidence of available clinical communication tools in this context and their effectiveness.
If the author wishes to add an ethical argument (which certainly augments the novelty of the paper), I would suggest he invite a moral philosopher or theologian to co-write the piece and help set it out more clearly.
Reviewer 2 Report
I would like to thank the author for the precious reflection made by taking inspiration from the experience of COVID- 19, but which touches the work of many professionals who takes care of patients on a daily basis.
I find the author's reflections deepened and supported by extensive literature.
The suggestions provided can concretely assist physicians and patients during their decision making.
Two small revisions:
- line132 I think there was an error in the reference (it would sound to be the 6 and not the 7 )
- line141 there is a repetition in the text ( ..to we need..).
Thanks for your thoughts.
Round 2
Reviewer 1 Report
The new draft is not very different from the first and does not represent the major re-write I envisaged in my last review. There are some areas where the relatively small changes the author has made have resulted in an improvement.
The most obvious is that the article's focus is now more clearly on the importance of managing uncertainty in medical practice, rather than on ethics. The author's point is that an emphasis on resolving complex decisions into binary outcomes (ventilate/do not ventilate) misleads patients and as such paradoxically makes them less able to make informed choices. He proposes ASIPP as a way of avoiding that inadvertent incapacitation.
Several factors identify this as an opinion piece rather than a review. The title suggests the piece is about COVID but after the first few paragraphs the author moves without explanation to talking about ICU more generally. The language is often informal, with several anecdotes using the first person singular. He evokes little evidence to support his contention that false certainties incapacitate patients (though I think he is right). The logic is poor (the distinction between 'people' and 'patients' in line 224/225, for example, is bewildering) with the result that the argument is not convincing. While it is true to say that predicting certainties in medicine is a form of dishonesty, it does not follow that the only honest response is that we cannot predict anything. I do not know for certain that a certain adult with COVID will not recover, but it would not be true to say that I know nothing about how likely it is that she will survive. I can definitely know that it is unlikely. It is not clear to me that the author acknowledges that such an honest form of certainty can exist.
In summary, this is a piece that expresses a strongly held opinion on the part of its author. It is an opinion I and many others largely share and would be happy to see promulgated. I am not sure this piece represents a convincing account of it.
Author Response
Thank you for these comments. Yes, this is more of a opinion piece or commentary than a systematic review and was submitted as such. COVID is not in the title but is in the introduction as an topical example of serious illness, which is what this paper is about. As I have stated before, as a commentary, I take literary licence to offer my opinions and anecdotes. The paper is full of citations that provide evidence for my assertions. The reference to people vs patients is intentional. I decry ‘people’ doing advance care planning at their kitchen table making decisions about their future medical care and am promoting the idea that ‘patients’ work with clinicians to make informed treatment decisions in the moment. I am not saying that doctors can’t provide probabilistic prognostic statements, I am saying that people shouldn’t make fixed medical care plans for death when death is uncertain but should rather prepare to work with doctors, who can prognosticate in the context of serious illness, to develop the treatment plan that is right for them. I think these points are fairly clear and no further revisions are needed.